# Neutrophil infiltration regulates clock-gene expression to organize daily hepatic metabolism

María Crespo[1†], Barbara Gonzalez-Teran[1†], Ivana Nikolic[1], Alfonso Mora[1], Cintia Folgueira[1], Elena Rodríguez[1], Luis Leiva-Vega[1], Aránzazu Pintor-Chocano[1], Macarena Fernández-Chacón[1], Irene Ruiz-Garrido[1], Beatriz Cicuéndez[1], Antonia Tomás-Loba[1], Noelia A-Gonzalez[1], Ainoa Caballero-Molano[1], Daniel Beiroa[2,3], Lourdes Hernández-Cosido[4], Jorge L Torres[5], Norman J Kennedy[6], Roger J Davis[6], Rui Benedito[1], Miguel Marcos[5], Ruben Nogueiras[2,3], Andrés Hidalgo[1], Nuria Matesanz[1*], Magdalena Leiva[1†*], Guadalupe Sabio[1*]

[1]Centro Nacional de Investigaciones Cardiovasculares Carlos (CNIC), Madrid, Spain; [2]CIBER Fisiopatología de la Obesidad y Nutrición (CIBERobn), Santiago de Compostela, Spain; [3]CIMUS, University of Santiago de Compostela-Instituto de Investigación Sanitaria, Santiago de Compostela, Spain; [4]Department of General Surgery, University Hospital of Salamanca-IBSAL, Department of Surgery, University of Salamanca, Salamanca, Spain; [5]Department of Internal Medicine, University Hospital of Salamanca-IBSAL, Department of Medicine, University of Salamanca, Salamanca, Spain; [6]Howard Hughes Medical Institute and Program in Molecular Medicine, University of Massachusetts Medical School, Worcester, United States

*For correspondence:
nuria.matesanz@cnic.es (NM);
magdalena.leiva@cnic.es (ML);
gsabio@cnic.es (GS)

[†]These authors contributed equally to this work

Competing interests: The authors declare that no competing interests exist.

**Abstract** Liver metabolism follows diurnal fluctuations through the modulation of molecular clock genes. Disruption of this molecular clock can result in metabolic disease but its potential regulation by immune cells remains unexplored. Here, we demonstrated that in steady state, neutrophils infiltrated the mouse liver following a circadian pattern and regulated hepatocyte clock-genes by neutrophil elastase (NE) secretion. NE signals through c-Jun NH2-terminal kinase (JNK) inhibiting fibroblast growth factor 21 (FGF21) and activating *Bmal1* expression in the hepatocyte. Interestingly, mice with neutropenia, defective neutrophil infiltration or lacking elastase were protected against steatosis correlating with lower JNK activation, reduced *Bmal1* and increased FGF21 expression, together with decreased lipogenesis in the liver. Lastly, using a cohort of human samples we found a direct correlation between JNK activation, NE levels and *Bmal1* expression in the liver. This study demonstrates that neutrophils contribute to the maintenance of daily hepatic homeostasis through the regulation of the NE/JNK/*Bmal1* axis.

## Introduction

Circadian rhythms regulate several biological processes through internal molecular mechanisms (*Dibner et al., 2010*) and the chronic perturbation of circadian rhythms is associated with the appearance of metabolic syndrome (*Kolla and Auger, 2011*). This homeostasis is closely dependent on the circadian system in the liver, which shows rhythmic expression of enzymes associated with glucose and lipid metabolism (*Haus and Halberg, 1966*; *North et al., 1981*; *Tahara and Shibata, 2016*). Moreover, mice with mutations in clock genes encoding nuclear receptors have impaired glucose and lipid metabolism and are susceptible to diet-induced obesity and metabolic dysfunction,

**eLife digest** Every day, the body's biological processes work to an internal clock known as the circadian rhythm. This rhythm is controlled by 'clock genes' that are switched on or off by daily physical and environmental cues, such as changes in light levels. These daily rhythms are very finely tuned, and disturbances can lead to serious health problems, such as diabetes or high blood pressure.

The ability of the body to cycle through the circadian rhythm each day is heavily influenced by the clock of one key organ: the liver. This organ plays a critical role in converting food and drink into energy. There is evidence that neutrophils – white blood cells that protect the body by being the first response to inflammation – can influence how the liver performs its role in obese people, by for example, releasing a protein called elastase. Additionally, the levels of neutrophils circulating in the blood change following a daily pattern. Crespo, González-Terán et al. wondered whether neutrophils enter the liver at specific times of the day to control liver's daily rhythm.

Crespo, González-Terán et al. revealed that neutrophils visit the liver in a pattern that peaks when it gets light and dips when it gets dark by counting the number of neutrophils in the livers of mice at different times of the day. During these visits, neutrophils secreted elastase, which activated a protein called JNK in the cells of the mice's liver. This subsequently blocked the activity of another protein, FGF21, which led to the activation of the genes that allow cells to make fat molecules for storage. JNK activation also switched on the clock gene, *Bmal1*, ultimately causing fat to build up in the mice's liver. Crespo, González-Terán et al. also found that, in samples from human livers, the levels of elastase, the activity of JNK, and whether the *Bmal1* gene was switched on were tightly linked. This suggests that neutrophils may be controlling the liver's rhythm in humans the same way they do in mice.

Overall, this research shows that neutrophils can control and reset the liver's daily rhythm using a precisely co-ordinated series of molecular changes. These insights into the liver's molecular clock suggest that elastase, JNK and *Bmal1* may represent new therapeutic targets for drugs or smart medicines to treat metabolic diseases such as diabetes or high blood pressure.

consistent with the idea that these genes control hepatic metabolic homeostasis (*Delezie et al., 2012*; *Kudo et al., 2008*; *Lamia et al., 2008*; *Rey et al., 2011*; *Tong and Yin, 2013*; *Turek et al., 2005*; *Yang et al., 2006*). Besides, recent reports have shown that hepatic physiology follows a diurnal rhythm driven by clock genes, with expression of proteins involved in fatty acid synthesis higher in the morning while those controlling fatty acid oxidation are higher at sunset (*Toledo et al., 2018*; *Zhou et al., 2015*).

Blood leukocyte levels also oscillate diurnally, as does the release of hematopoietic stem cells and progenitor cells from the bone marrow (BM) (*Haus and Smolensky, 1999*; *Lucas et al., 2008*; *Méndez-Ferrer et al., 2008*) and their recruitment into tissues (*Adrover et al., 2019*; *He et al., 2018*; *Scheiermann et al., 2012*). Oscillatory expression of clock genes in peripheral tissues is largely tuned by the suprachiasmatic nucleus (*Dibner et al., 2010*; *Druzd and Scheiermann, 2013*; *Huang et al., 2011*; *Reppert and Weaver, 2002*); however, the potential regulation of daily rhythms of specific tissues by immune cells remains largely unexplored, both in steady state and during inflammation. Although the molecular mechanisms linking circadian rhythms and metabolic disease are largely unknown, several studies have demonstrated a strong association between leukocyte activation and metabolic diseases (*McNelis and Olefsky, 2014*). A prime example is the BM, where engulfment of infiltrating neutrophils by tissue-resident macrophages modulates the hematopoietic niche (*Casanova-Acebes et al., 2013*).

The circadian clock is dysregulated by obesity (*Kohsaka et al., 2007*; *Xu et al., 2014*), and recent studies suggest that liver leukocyte recruitment and migration show a circadian rhythm (*Scheiermann et al., 2012*; *Solt et al., 2012*) whose alteration can result in steatosis (*Solt et al., 2012*; *Xu et al., 2014*). Neutrophils are key factors in steatosis development (*González-Terán et al., 2016*; *Keller et al., 2009*; *Mansuy-Aubert et al., 2013*; *Nathan, 2006*) and show diurnal oscillations in their recruitment and migration to multiple tissues (*Scheiermann et al., 2012*; *Solt et al., 2012*). Here, we demonstrate that circadian neutrophil infiltration into the liver controls the expression of

clock genes through the regulation of c-Jun NH2-terminal kinase (JNK) and the hepatokine fibroblast growth factor 21 (FGF21), driving adaptation to daily metabolic rhythm.

## Results

### Rhythmic neutrophil infiltration into the liver modulates the expression of hepatic clock genes

Virtually all cell types have an internal clock that controls their rhythmicity through the periodic expression of clock genes (*Robles et al., 2014*; *Tahara and Shibata, 2016*). However, it is unknown how these multiple cell rhythms are integrated. The liver is an essential metabolic organ that controls body glucose and lipid homeostasis (*Manieri and Sabio, 2015*), and neutrophil infiltration alters its function (*González-Terán et al., 2016*). We hypothesized that the metabolic cycles in the liver might be entrained by rhythmic neutrophil infiltration. To test this, we harvested liver, BM, and blood from C57BL6J mice at 4 hr intervals over a 24 hr period. Liver neutrophil infiltration showed a clear diurnal pattern, with a peak at ZT2, coinciding with liver-driven lipogenesis in mice (*Zhou et al., 2015*), and a nadir during the night, at ZT14 (*Figure 1A*), correlating with lipolysis (*Zhou et al., 2015*). These oscillations corresponded directly to changes in neutrophil numbers in blood (*Figure 1—figure supplement 1A*), suggesting that liver infiltration might result from higher neutrophil migration to the liver. We first confirmed that neutrophils were infiltrated in the liver using 3D microscopy. According to published data (*Casanova-Acebes et al., 2018*), infiltrated neutrophils presented an intrasinusoidal distribution in the liver, different to that observed in the Kupffer cells population (*Figure 1B* and *Figure 1—figure supplement 1B*). Then we evaluated whether myeloid chemokines could be involved in circadian neutrophil recruitment into the liver. Analysis of liver lysates indicated that the expression of the hepatocyte-derived neutrophil chemoattractant *Cxcl1* (*Su et al., 2018*) was higher at ZT2 than a ZT14. Moreover, mRNA of *Cxcl1* in liver samples showed the same oscillation pattern than infiltrated neutrophils, suggesting that this chemokine may be important in the regulation of the neutrophil diurnal cycle (*Figure 1—figure supplement 1C*).

The infiltration pattern correlated with liver expression levels of the clock-gene *Bmal1*, peaking at ZT2 and bottoming at ZT14 (*Figure 1C*). Infiltration also correlated inversely with the expression of *Nr1d2* (encoding Rev-erb β), *Per2*, and *Cry2* (*Figure 1C*), which are important proteins in the control of circadian rhythms (*Reppert and Weaver, 2002*), consistent with the feedback loop that controls their expression. *Bmal1* is thought to induce lipogenesis (*Zhang et al., 2014*), whereas *Nr1d2* controls lipid metabolism and its reduced expression promotes lipogenesis and steatosis (*Delezie et al., 2012*; *Solt et al., 2012*). In agreement with these studies, liver triglycerides were higher at ZT2 than at ZT14 (*Figure 1D*).

Our results show a correlation between neutrophil infiltration, hepatocyte *Bmal1* expression, and lipid metabolism regulation, raising the possibility that neutrophils signal to hepatocytes to modulate the expression of circadian genes. Exposure of mouse hepatocytes *in vitro* to freshly isolated neutrophils increased hepatocyte expression of the clock genes *Bmal1* and *Clock*. In contrast, no effect was observed upon exposure to T or B lymphocytes, or macrophages, suggesting the existence of a neutrophil-to-hepatocyte communication that controls hepatocyte clock-gene expression (*Figure 1E* and *Figure 1—figure supplement 1D*).

We then investigated whether neutrophil elastase (NE), a proteolytic enzyme reported to regulate liver metabolism, could regulate hepatocyte clock genes (*Mansuy-Aubert et al., 2013*; *Talukdar et al., 2012*). Exposure to elastase reproduced the same increase in hepatocyte *Bmal1* and *Clock* expression in contrast with another protease that did not affect *Bmal1* expression (*Figure 1F* and *Figure 1—figure supplement 1D*).

Next, neutrophil-mediated regulation of liver clock-gene expression *in vivo* was investigated using a previously characterized genetic model of neutrophil deficiency (*Dzhagalov et al., 2007*; *Steimer et al., 2009*; *Figure 1—figure supplement 1E,F* and *Figure 1—figure supplement 2A–C*). Low hepatic neutrophil infiltration in neutropenic mice correlated with reduced expression of *Bmal1* and *Clock* (*Figure 1G*) and increased expression of *Cry2* and *Per2* at ZT2 (*Figure 1G*). These changes in clock-gene expression were accompanied by lower liver triglyceride levels (*Figure 1H*). Furthermore, lack of neutrophils perturbed the diurnal rhythmicity in *Bmal1*, *Clock*, and *Per2* expression in the liver without affecting clock genes in other organs such as the lung, in which there is no

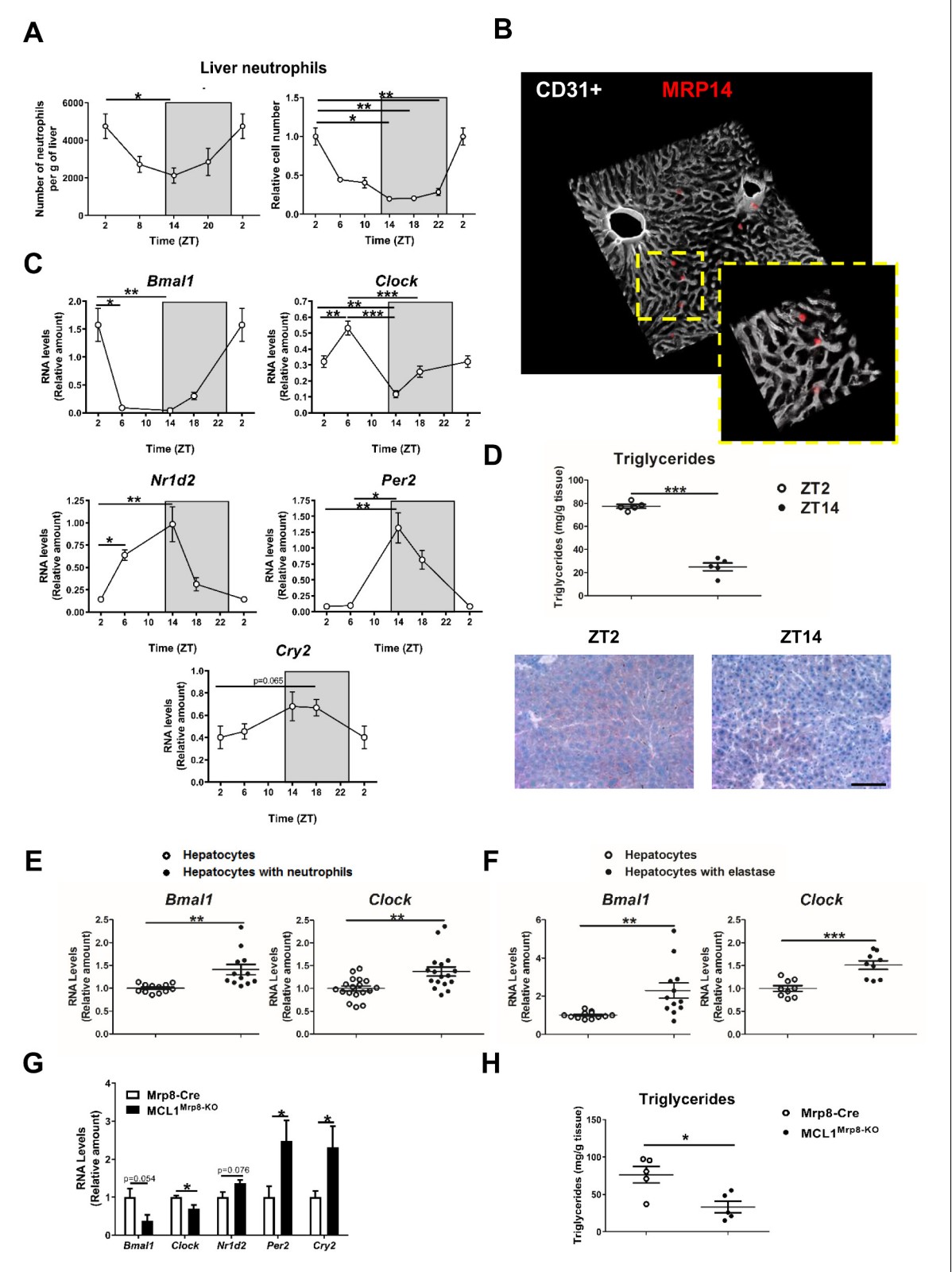

**Figure 1.** Neutrophil infiltration into the liver controls hepatic clock-gene expression. (**A**) Flow cytometry analysis of the CD11b⁺Ly6G⁺ liver myeloid subset, isolated from C57BL6J mice at the indicated ZTs. Left, CD11b⁺Ly6G⁺ liver myeloid subset analyzed at 6 hr intervals and normalized by the tissue weight. Right, percentage of CD11b⁺Ly6G⁺ population analyzed at 4 hr intervals and normalized to ZT2 (n = 5). (**B**) Representative 3-D image of liver section showing the distribution on infiltrated neutrophils. Livers were stained with anti-S100A9 (Mrp14) (red) and vessels were stained with anti-

*Figure 1 continued on next page*

*Figure 1 continued*

CD31 and anti-endomucin (grey). Sizes of the liver sections are 510 x 510 x 28 µm and 160 x 160 x 28 µm, respectively. (C) qRT-PCR analysis of circadian clock-gene and nuclear-receptor mRNA expression in livers from C57BL6J mice at the indicated ZTs (n = 5). (D) Liver triglycerides and oil-red-stained liver sections prepared from C57BL6J mice at ZT2 and ZT14. Scale bar, 50 µm (n = 5). (E) qRT-PCR analysis of clock-gene mRNA in hepatocyte cultures exposed to freshly isolated FMLP-activated neutrophils (n = 4-6 wells of 3 independent experiments). (F) qRT-PCR analysis of clock-gene mRNA in hepatocyte cultures treated with 5 nM elastase (n = 3-4 wells of 3 independent experiments). (G) qRT-PCR analysis of clock-gene and nuclear-receptor mRNA expression in livers from control mice (Mrp8-Cre) and neutropenic mice (MCL1$^{Mrp8-KO}$) sacrificed at ZT2 (n = 5). (H) Hepatic triglycerides detected in livers from control mice (Mrp8-Cre) and neutropenic mice (MCL1$^{Mrp8-KO}$) at ZT2 (n = 5). Data are means ± SEM from at least 2 independent experiments. *$p<0.05$; **$p<0.01$; ***$p<0.005$ (A, left panel) One-way ANOVA with Tukey's post hoc test. (A, right panel) Kruskal-Wallis test with Dunn's post hoc test. (C) One-way ANOVA with Tukey's post hoc test or Kruskal-Wallis test with Dunn's post hoc test. (D to H) *t*-test or Welch's test. ZT2 point is double plotted to facilitate viewing.

The online version of this article includes the following source data and figure supplement(s) for figure 1:

**Source data 1.** Raw data and statistical test.
**Figure supplement 1.** Neutrophils follow a circadian rhythm.
**Figure supplement 1—source data 1.** Raw data and statistical test.
**Figure supplement 2.** Neutrophil deficiency alters clock-gene expression.
**Figure supplement 2—source data 1.** Raw data and statistical test.

correlation between the peak of neutrophil infiltration and *Bmal1* expression (*Figure 1—figure supplement 2D,E*). Our results thus indicate that neutrophils might specifically control the expression of hepatocyte circadian clock genes in steady state.

## Disruption of daily neutrophil infiltration in the liver affects hepatocyte molecular clock and metabolism

Chronic jet lag alters liver circadian genes and disrupts liver metabolism (*Kettner et al., 2016*). Analysis of a mouse model of jet lag revealed complete disruption of the circadian liver neutrophil infiltration with increased hepatic neutrophil infiltration even at ZT14 (*Figure 2A*). Abolition of rhythmic neutrophil hepatic infiltration under jet lag correlated with increased steatosis and high levels of liver triglycerides (*Figure 2B*). To evaluate whether the metabolic effect of circadian perturbation was caused by the increased neutrophil infiltration, we exposed neutropenic and control mice to the jet lag protocol (*Figure 2—figure supplement 1A,B*). Jet lag-induced steatosis was less severe in neutropenic mice (*Figure 2C*), and disruption of diurnal liver expression of *Bmal1* detected in control jet-lagged mice was partially ablated in neutropenic mice (*Figure 2D*). Similar results were also observed in mice with impaired neutrophil migration such as Cxcr2$^{MRP8-KO}$ BM transplanted mice (*Eash et al., 2010*; *Mei et al., 2012*) and p38γ/δ$^{Lyzs-KO}$ mice (*González-Terán et al., 2016*). In both models, the reduction of neutrophil infiltration correlated with decreased levels of liver *Bmal1* expression and protection from jet lag-induced steatosis (*Figure 2—figure supplement 1C–G*). These results are consistent with the role of neutrophils in the control of liver clock genes.

Inflammation plays a key role in the pathogenesis of non-alcoholic fatty liver disease (*Tiniakos et al., 2010*) and the development of hepatic steatosis is associated with increased liver infiltration by myeloid cells, particularly neutrophils (*González-Terán et al., 2016*; *Mansuy-Aubert et al., 2013*; *Talukdar et al., 2012*; *Tiniakos et al., 2010*). Two widely used mouse models of hepatic steatosis, high-fat diet (HFD) and methionine-choline-deficient (MCD) diet, increased liver neutrophil infiltration in WT mice at ZT2, ZT14, and ZT18 (*Figure 2E,F*). Consistent with a neutrophil-to-hepatocyte communication in the regulation of hepatocyte clock genes, the MCD diet enhanced *Bmal1* expression and inhibited *Cry2* and *Per2* expression in control mice, but not in neutropenic mice at ZT2 (*Figure 2G*). Altered liver clock-gene regulation in neutropenic mice was associated with protection against steatosis and lower liver triglycerides (*Figure 2H*). To confirm the role of neutrophils in modulating liver clock genes, we depleted neutrophils by injecting anti-Ly6G antibody into MCD diet-fed mice (*González-Terán et al., 2016*). Anti-Ly6G administration for 7 days reduced circulating neutrophil levels without affecting monocytes (*Figure 2—figure supplement 2A,B*), and treatment for 21 days markedly decreased hepatic diurnal *Bmal1* and *Clock* expression, increased expression of *Cry2*, and *Per2* (*Figure 2—figure supplement 2C*) and consequently reduced steatosis (*González-Terán et al., 2016*).

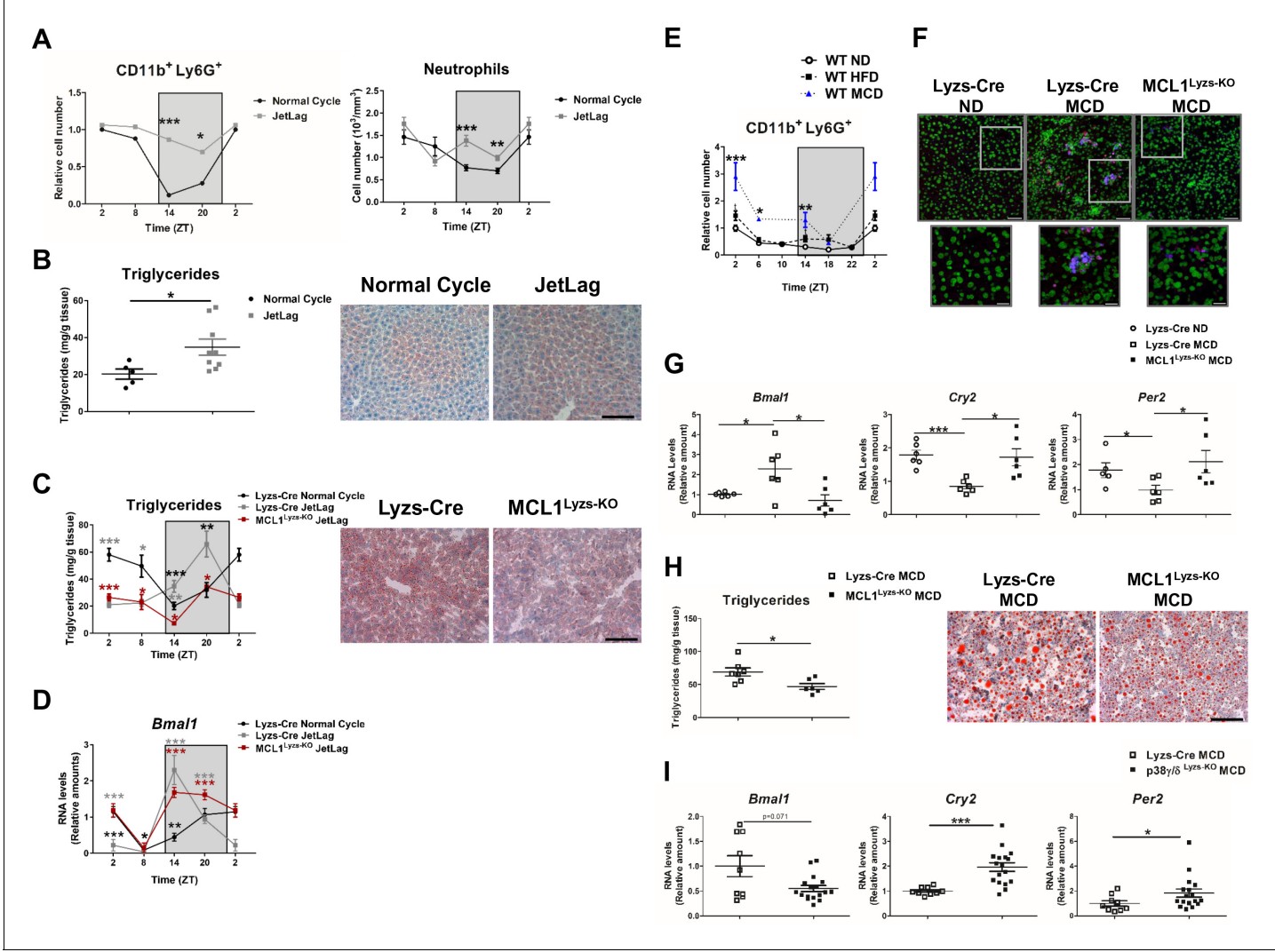

**Figure 2.** Increased hepatic neutrophil infiltration alters clock-genes expression and augments triglyceride content in the liver. (A–D) Control (Lyzs-Cre) (A–B) and control and neutropenic (MCL1[Lyzs-KO]) mice (C–D) were housed for 3 weeks with a normal 12 hr: 12 hr light/dark cycle (Normal Cycle) or with the dark period extended by 12 hr every 5 days (JetLag). Samples were obtained at the indicated ZTs. (A) Left, flow cytometry analysis of the CD11b[+]Ly6G[+] liver myeloid subset. Data represents the percentage CD11b[+]Ly6G[+] normalized to Normal Cycle ZT2. Right, circulating neutrophils in whole blood. (n = 5-8). (B) Liver triglycerides and representative oil-red-stained liver sections at ZT14. Scale bar, 50 μm (n = 9-10). (C) Hepatic triglyceride content analyzed at 6 hr intervals, and representative oil-red-stained liver sections at ZT14. Scale bar, 50 μm (n = 4-6). (D) qRT-PCR analysis of *Bmal1* mRNA in livers. (n = 5-8). (E) Flow cytometry analysis of the CD11b[+]Ly6G[+] liver myeloid subset isolated at 6 hr intervals from C57BL6J mice fed a ND, a HFD (8 weeks) or a MCD (3 weeks). The chart shows the CD11b[+]Ly6G[+] population as a percentage of the total intrahepatic CD11b[+] leukocyte population normalized to ND group at ZT2 (n = 5 to 10). (F–I) Control mice (Lyzs-Cre) and neutropenic mice (MCL1[Lyzs-KO]) or p38γ/δ[Lyzs-KO] were fed a ND or the MCD diet for 3 weeks and sacrificed at ZT2. (F) Representative images of the infiltration of neutrophils in the liver stained with anti-Mrp14 (blue) and anti-NE (red); nuclei with Sytox Green. Scale bar, 50 μm (Top) and 25 μm (Bottom). (G) qRT-PCR analysis of clock-gene expression in livers (n = 6). (H) Liver triglycerides and representative oil-red-stained liver sections. Scale bar, 50 μm (n = 7-6). (I) qRT-PCR analysis of clock genes in livers at ZT2 (n = 9-17). Data are means ± SEM from at least two independent experiments. *p<0.05; **p<0.01; ***p<0.005 (A to D) *t-test* or Welch's test. (E) Two-way ANOVA with Fisher's post hoc test; p<0.05 ND vs HFD; p<0.0001 ND vs MCD. *p<0.05; ***p<0.005 (G to I) *t-test* or Welch's test. ZT2 point is double plotted to facilitate viewing.

The online version of this article includes the following source data and figure supplement(s) for figure 2:

**Source data 1.** Raw data and statistical test.
**Figure supplement 1.** Defective neutrophil migration to the liver alters hepatic clock- gene expression and triglyceride content.
**Figure supplement 1—source data 1.** Raw data and statistical test.
**Figure supplement 2.** Neutrophil depletion alters hepatic clock-gene expression.
**Figure supplement 2—source data 1.** Raw data and statistical test.
**Figure supplement 2—source data 2.** Raw data and statistical test.

To further support the role of neutrophil liver infiltration in the regulation of liver clock genes and hepatic lipogenesis during diet-induced steatosis, we leveraged a mouse model (p38γ/δ$^{Lyzs-KO}$) that exhibits deficient neutrophil migration and subsequently, reduced liver neutrophil infiltration after MCD diet (*González-Terán et al., 2016*). Compared with diet-matched control (Lyzs-Cre) mice, MCD-diet-fed p38γ/δ$^{Lyzs-KO}$ mice showed hepatic down-regulation of *Bmal1*, which was associated with higher expression of *Cry2*, and *Per2* (*Figure 2I*). These results suggest that the reduced neutrophil infiltration in mice lacking myeloid p38γ/δ expression is responsible for the altered expression of circadian clock genes. Overall, these findings strongly support that neutrophil infiltration modulates clock-gene expression in the liver, with downstream effects on liver metabolism.

## Regulation of daily hepatic metabolism by neutrophils through JNK-FGF21 axis

It has been suggested that JNK activation in the liver may be regulated in a circadian manner with a peak at noon (*Robles et al., 2014*). To evaluate whether neutrophils might mediate this diurnal regulation of JNK, we analyzed JNK activation in neutropenic mice. Lack of neutrophils was associated with lower liver expression and activation of JNK, lower activation of the JNK downstream effector c-Jun, and lower expression of acetyl-CoA carboxylase (*Acaca*), a key enzyme in metabolic regulation (acetyl-CoA carboxylase; ACC) that mediates inhibition of beta-oxidation and activation of lipid biosynthesis (*Figure 3A* and *Figure 3—figure supplement 1A*). Similar results were found in p38γ/δ$^{Lyzs-KO}$ mice, in which reduced liver neutrophil infiltration was associated with decreased JNK phosphorylation and ACC protein levels (*Figure 3B* and *Figure 3—figure supplement 1B*). Moreover, neutrophil-treated hepatocytes showed increased JNK activation together with increased levels of ACC expression (*Figure 3—figure supplement 1C*). NE represents a potential mediator of this neutrophil function because elastase-treated hepatocytes also showed higher JNK activation, suggesting that this protease modulates the expression of the clock genes through the JNK signaling pathways (*Figure 3C* and *Figure 3—figure supplement 1D*). This JNK activation was accompanied by increased *Bmal1* expression (*Figure 3D*), indicating that neutrophils altered liver clock-gene expression through the elastase-JNK pathway.

Our results suggest that neutrophil-mediated JNK activation might modulate hepatocyte clock genes and metabolism through the regulation of ACC. Supporting this hypothesis, specific JNK depletion in hepatocytes downregulated *Bmal1*, *Clock*, and *Acaca* compared to Alb-Cre (*Figure 3E* and *Figure 3—figure supplement 1E*). According to these results, JNK inhibition reduced the expression of *Bmal1*, *Clock* and *Acaca* in WT liver but not in neutropenic mice (*Figure 3—figure supplement 1F,G*). These data strongly suggest that JNK activation caused by neutrophil infiltration modulates clock genes and daily metabolism in hepatocytes.

JNK is an important modulator of the expression of the hepatokine circadian regulator FGF21 (*Vernia et al., 2014*), which controls glucose and lipid metabolism (*Fisher and Maratos-Flier, 2013*; *Li et al., 2013*; *Potthoff et al., 2012*). Mice lacking JNK in hepatocytes had higher FGF21 mRNA expression (*Figure 3E*). In concordance with high JNK activation, FGF21 expression was reduced in neutrophil-exposed hepatocytes (*Figure 3—figure supplement 1H*). Moreover, neutropenic and p38γ/δ$^{Lyzs-KO}$ mice showed increased FGF21 expression (*Figure 3F* and *Figure 3—figure supplement 1I,J*), which was consistent with the reduced hepatocyte JNK activation in these mice.

To further define the role of FGF21 in the neutrophil-mediated regulation of liver metabolism, we suppressed FGF21 expression using two independent lentiviral shRNA vectors (*Figure 3G* and *Figure 3—figure supplement 1K*). The protection of p38γ/δ$^{Lyzs-KO}$ mice against MCD-diet-induced alterations was abrogated by shFGF21 and these mice developed steatosis with an elevated hepatic triglyceride content (*Figure 3H,I*). These data further supported the idea that neutrophil infiltration controls liver metabolism through the regulation of FGF21 expression.

## Neutrophil elastase deficiency affects the expression patterns of clock genes and lipid metabolism

To formally confirm the involvement of NE in circadian clock alteration, we first evaluated the diurnal oscillation of NE levels in liver from WT mice fed a normal diet (ND). According to infiltration pattern of neutrophils in the liver (*Figure 1A*), we found higher NE levels at ZT2 than at ZT14. (*Figure 4A*). Next, circadian clock-gene expression in NE$^{-/-}$ mice revealed lower *Bmal1* and elevated *Per2* and

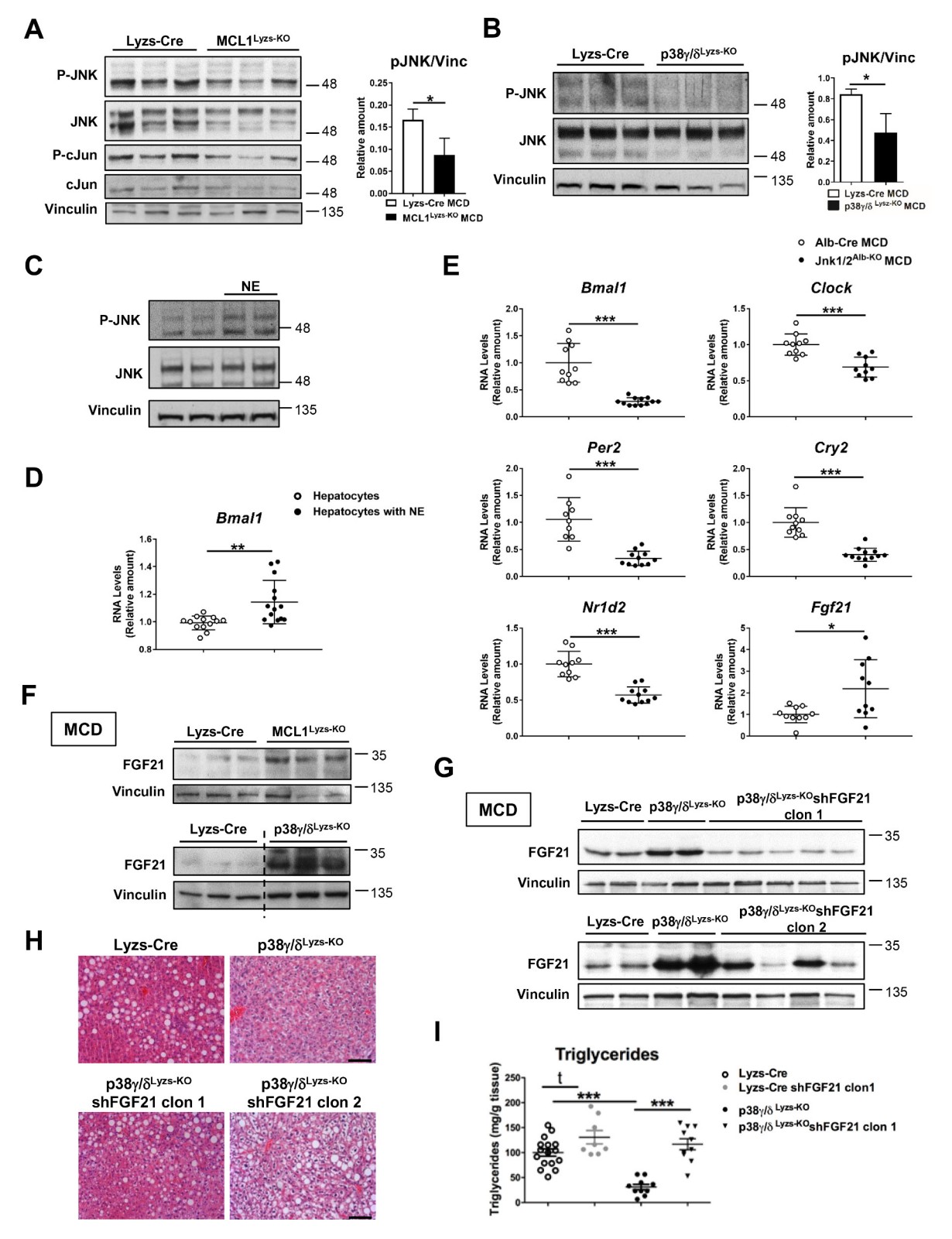

**Figure 3.** Diurnal regulation of liver metabolism involves neutrophil-mediated regulation of JNK and the hepatokine FGF21. Immunoblot analysis of JNK content and activation at ZT2 in liver extracts prepared from control (Lyzs-Cre) and neutropenic (MCL1[Lyzs-KO]) mice fed a MCD diet for 3 weeks (**A**) or Lyzs-Cre and p38γ/δ[Lyzs-KO] mice after 3 weeks of MCD diet (**B**). Immunoblot analysis of JNK content and activation (**C**) and *Bmal1* RNA expression (**D**) in hepatocyte cultures exposed to NE for 2 hr (n = 14 wells of 3 independent experiments). Immunoblot quantification is shown in *Figure 3—figure*

*Figure 3 continued on next page*

*Figure 3 continued*

supplement 1D (E) qRT-PCR analysis of clock genes and *Fgf21* in livers from Alb-Cre, and JNK1/2$^{Alb-KO}$ mice after 3 weeks of MCD diet at ZT2 (n = 9-12). (F) Immunoblot analysis of FGF21 content in liver extracts prepared from control (Lyzs-Cre) and neutropenic (MCL1$^{Lyzs-KO}$) mice, or from Lyzs-Cre, and p38γ/δ$^{Lyzs-KO}$ mice after 3 weeks of MCD diet sacrificed at ZT2. Immunoblot quantification is shown in *Figure 3—figure supplement 1I,J*. (G–I) Lyzs-Cre and p38γ/δ$^{Lyzs-KO}$ mice were injected with 2 shRNA independent clones targeting FGF21. Seven days after infection, mice were placed on the MCD diet and sacrificed after 3 weeks at ZT2. (G) Immunoblot analysis of FGF21 content in liver extracts prepared from Lyzs-Cre, p38γ/δ$^{Lyzs-KO}$, and p38γ/δ$^{Lyzs-KO}$ mice infected with FGF21 shRNA. Immunoblot quantification is shown in *Figure 3—figure supplement 1K*. (H) Representative H&E-stained liver sections. Scale bar, 50 µm. (I) Hepatic triglyceride content at the end of the treatment period (n = 8-10). Data are means ± SEM from at least 2 independent experiments. *p<0.05; **p<0.01; ***p<0.005 (A, B, D and E) *t*-test or Welch's test. (I) One-way ANOVA with Bonferroni post hoc test or *t*-test.

The online version of this article includes the following source data and figure supplement(s) for figure 3:

**Source data 1.** Raw data and statistical test.
**Figure supplement 1.** Neutrophils regulate hepatic metabolism and clock genes through JNK and FGF21.
**Figure supplement 1—source data 1.** Raw data and statistical test.

---

*Cry2* expression, compared to control mice (*Figure 4B*), which mimicked the behavior of neutropenic mice. In addition, NE$^{-/-}$ mice presented lower respiratory quotient during the lights-on period than WT mice, indicating that these mice have increased fat utilization as a source of energy (*Figure 4C*), supporting the data that reduced liver-neutrophil infiltration results in higher lipid oxidation. Interestingly, when fed MCD or HFD diet, NE$^{-/-}$ mice were protected against steatosis (*Figure 4D,E* and *Figure 4—figure supplement 1A,B*), presented lower JNK activation, and expressed less ACC than control mice (*Figure 4F,G* and *Figure 4—figure supplement 1D*). Besides, NE$^{-/-}$ mice were protected against alterations in clock-gene expression induced by MCD diet, presenting lower expression of *Bmal1* and higher of *Cry2* and *Per2* comparing to control mice at ZT2 (*Figure 4H*). Furthermore, under HFD, NE$^{-/-}$ mice were also refractory to these changes as these mice maintained a pattern of clock-gene expression similar to control mice in ND (*Figure 4—figure supplement 1E*).

To formally test a direct contribution of NE in the regulation of hepatic clock-gene expression and liver metabolism, we infused WT or NE$^{-/-}$ neutrophils into neutropenic mice under the jet lag protocol (*Figure 5A*). The infusion of WT neutrophils was able to increase *Bmal1* expression in the liver after jet lag, while neutropenic mice infused with NE$^{-/-}$ neutrophils presented the same levels of *Bmal1* than non-infused neutropenic mice (*Figure 5B*). In addition, while infusion of neutropenic mice with WT neutrophils increased steatosis, neutropenic mice infused with NE$^{-/-}$ neutrophils presented the same levels of steatosis than control neutropenic mice (*Figure 5C,D*). All these data indicate that diet or jet-lag -induced hepatic infiltration of neutrophils results in dysregulation of the liver clock, and the lack of NE is enough to protect mice against these alterations.

Finally, to evaluate the translational relevance of these findings for human physiology we quantified in human livers the expression levels for the genes encoding NE, *JUN* (as an indicator of JNK activation) and Bmal. Our results suggest that the levels of *ELANE* expression directly correlate with *BMAL1* and *JUN* mRNA in livers from a human cohort (*Figure 5E*). These correlations reinforce the idea that a rhythmic neutrophil infiltration in the liver controls the expression of clock genes through the JNK pathway activation and could be a target for therapeutic intervention during non-alcoholic fatty liver disease.

## Discussion

Our analysis demonstrates that neutrophils control clock genes in the liver and that reduced neutrophil infiltration protects against jet lag and diet-induced liver steatosis by altering the expression of these temporal regulators. These findings establish neutrophils as unexpected players in the regulation of daily hepatic metabolism. Our results also demonstrate that at least part of this neutrophil-induced clock modulation is mediated by elastase. These results agree with previous data showing that NE mediates the deleterious effects of neutrophils on liver metabolism and that mice lacking NE are protected against diet-induced steatosis (*Mansuy-Aubert et al., 2013*; *Talukdar et al., 2012*). The molecular mechanism underlying this regulation involves neutrophil NE that induces activation of JNK and consequently inhibits the production of the hepatokine FGF21. The JNK pathway

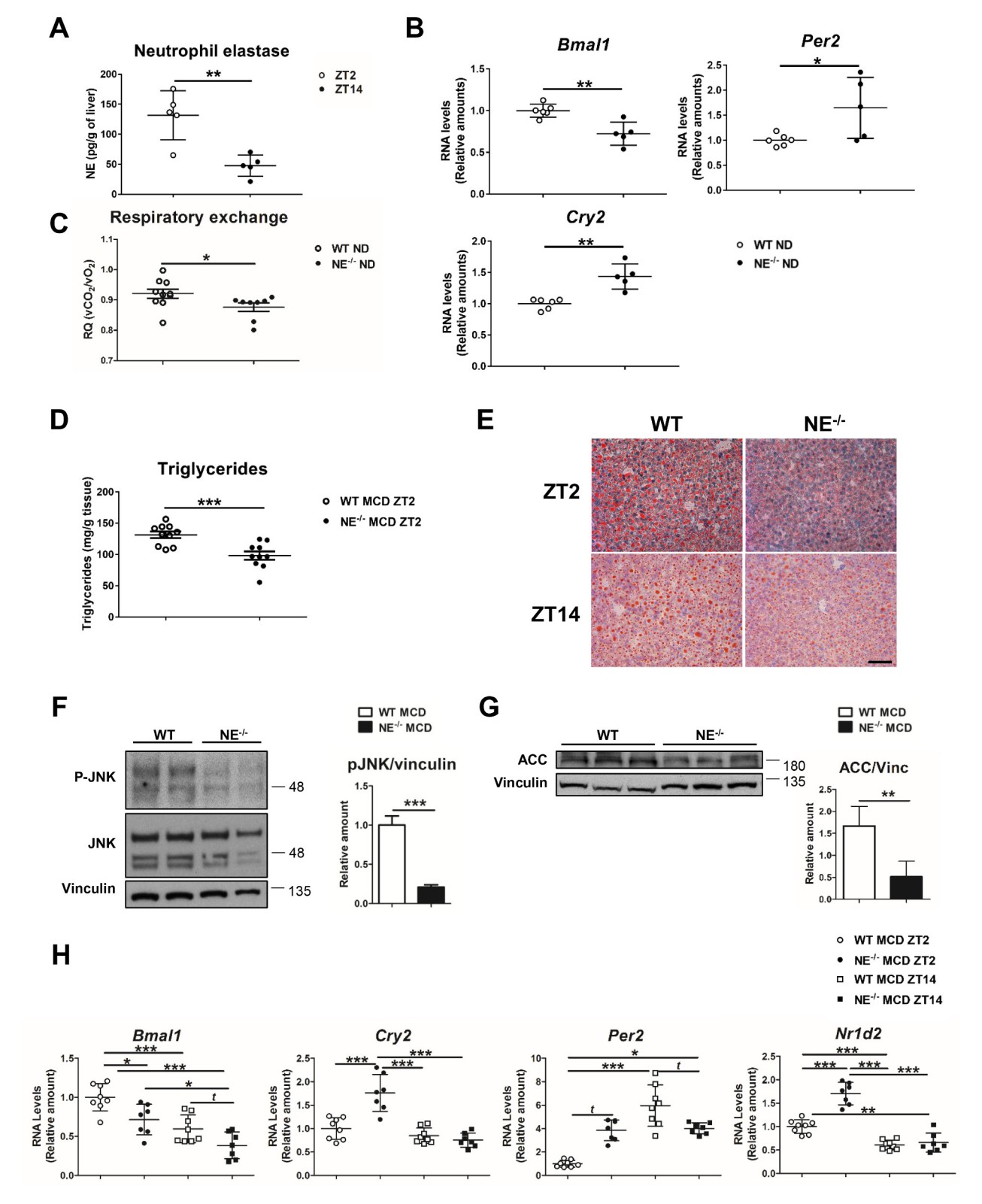

**Figure 4.** Elastase controls liver clock-gene expression modulating JNK activation. (**A**) Extracellular NE levels in livers from WT mice at ZT2 and ZT14. (**B**) qRT-PCR analysis of clock-genes and nuclear-receptor mRNA expression in livers from WT and NE KO mice (NE⁻/⁻) at ZT2 (n = 5–6). (**C**) Respiratory exchange ratio of WT and NE⁻/⁻ mice fed with ND. Results are from the lights-on period (n = 9). (**D–H**) WT and NE⁻/⁻ mice were fed a MCD diet for 3 weeks and sacrificed at the indicated time. (**D**) Liver triglycerides at the end of the diet period. (**E**) Representative oil-red-stained liver sections. Scale

*Figure 4 continued on next page*

*Figure 4 continued*

bar, 50 µm (n = 10). (F) Immunoblot analysis and quantifications of JNK content and activation in liver extracts prepared from WT and NE$^{-/-}$. (G) Immunoblot analysis and quantification of ACC content in liver extracts from WT and NE$^{-/-}$ mice. (H) qRT-PCR analysis of clock-genes and nuclear-receptor mRNA expression in livers from WT and NE$^{-/-}$ mice at ZT2 and ZT14 (n = 7–8). Data are means ± SEM from at least two independent experiments. *p<0.05; **p<0.01; ***p<0.005 (A to G) *t*-test or Welch's test. (H) One-way ANOVA with to Tukey's post hoc test, *t*-test or Welch's test.

The online version of this article includes the following source data and figure supplement(s) for figure 4:

**Source data 1.** Raw data and statistical test.

**Figure supplement 1.** Neutrophil elastase regulates daily hepatic metabolism through JNK.

is an important modulator of liver metabolism, and lack of JNK1 and JNK2 in hepatocytes protects against steatosis (*Manieri and Sabio, 2015*). Here, we also demonstrate that JNK also regulates hepatocyte clock genes and, therefore, modulates diurnal adaptation of liver metabolism.

Recently published data have demonstrated that lipogenesis is increased in the light phase, in agreement with our analysis (*Guan et al., 2018*). We show that neutrophil infiltration causes JNK activation down-stream of elastase secretion, a time-dependent process. Indeed, phosphoproteomic analysis of the hepatic phosphorylation network identifies JNK as a key signaling enzyme with peak activation at ZT6 (*Robles et al., 2017*) immediately prior to the peak of lipogenic gene expression (*Guan et al., 2018*). Our results suggest that neutrophils induce an accumulative activation of JNK with a peak during the day that would control the lipogenic program.

Recent evidence established that the metabolic effects of JNK in the liver are mediated by FGF21 (*Vernia et al., 2016*; *Vernia et al., 2014*). Our results now show that liver FGF21 expression can be modulated through the control of JNK by neutrophils. Reduction of FGF21 by shRNA reverted the protective effect and metabolic changes induced by reduced neutrophil infiltration. In conclusion, our results show that the diurnal oscillating migratory properties of neutrophils regulate liver function in a manner that preserves daily metabolic rhythms, and that disturbance of this rhythmicity can cause disease. These results might imply a novel mechanism of action for the potential use of clock-modulating small molecules in liver health.

## Materials and methods

### Study population

For the analysis of human liver mRNA levels, individuals were recruited among patients who underwent laparoscopic cholecystectomy for gallstone disease. The study was approved by the Ethics Committee of the University Hospital of Salamanca (Spain), and all subjects provided written informed consent to participate. Patients were excluded if they had a history of alcohol use disorders or excessive alcohol consumption, chronic hepatitis C or B, or body mass index ≥35. Baseline characteristics of these groups are listed in *Figure 5—source data 1*.

### Animal models

Neutropenic mice were generated with MCL1 (B6.129-Mcl1tm3Sjk/J) crossed with B6.Cg-Tg (S100A8-Cre,-EGFP)1Ilw/J mice or B6.129P2-Lyz2tm1(cre)Ifo/J mice. Mice deficient in NE, with compound JNK1/2 deficiency in hepatocytes, with Cxcr2 deficiency in neutrophils or with *p38γ/δ* deficiency in myeloid compartment have been described (*Belaaouaj et al., 1998*; *Das et al., 2011*; *Das et al., 2009*; *González-Terán et al., 2016*) All mice were backcrossed for 10 generations to the C57BL/6J background (Jackson Laboratory). Genotypes were confirmed by PCR analysis of genomic DNA.

Mice were housed under a 12 hr light:12 hr dark cycle (Light is on at Zeitgeber Time ZT0 and off at ZT12). For jet lag experiments, the 12 hr:12 hr dark/light cycle was disrupted by extending the dark cycle 12 hr every 5 days over 3 weeks (*Kettner et al., 2016*). Cxcr2$^{MRP8-KO}$ chimeras were generated by exposing WT recipient mice to 2 doses of ionizing radiation (625 Gy) and reconstituting them with 5 × 10$^6$ donor BM (Cxcr2$^{MRP8-KO}$) cells injected into the tail vein.

Mice were fed a methionine-choline-deficient (MCD) diet for 3 weeks or a high-fat diet (HFD) for 8 weeks (Research Diets Inc). For neutrophil depletion, mice mini-osmotic pumps (Alzet) were implanted with anti-Ly6G antibody or saline (0.4 mg/kg per day, 21 days). For JNK inhibition

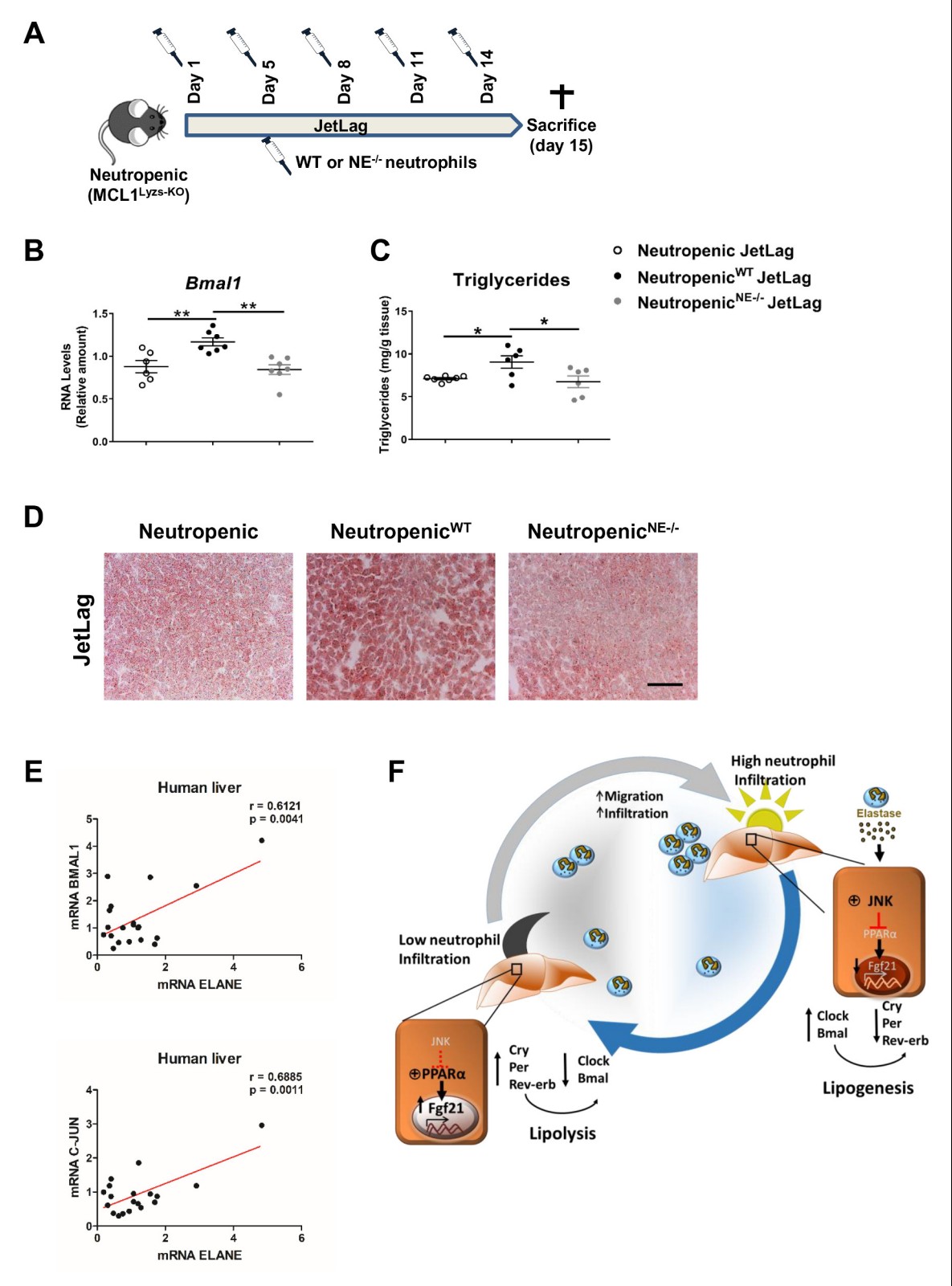

**Figure 5.** Neutrophil elastase reverses neutropenic mice phenotype through regulation of daily hepatic metabolism. (**A–D**) Neutropenic (MCL1$^{Lyzs-KO}$) mice were housed for 2 weeks with the dark period extended by 12 hr every 5 days (JetLag). Mice were infused with purified WT or NE$^{-/-}$ neutrophils. Samples were obtained at ZT14. (**A**) Picture describing the neutrophil infusion schedule during the JetLag protocol. (**B**) qRT-PCR analysis of *Bmal1* mRNA in livers. (**C**) Liver triglycerides and (**D**) representative oil-red-stained liver sections. Scale bar, 50 μm (n = 6-7). Data are means ± SEM. *p<0.05;
*Figure 5 continued on next page*

Figure 5 continued

t-test. (E) Correlation between mRNA levels of *BMAL1* and *ELANE* (r = 0.6141; p = 0.0052) or *JUN* and *ELANE* (r = 0.7362; p = 0.001105) in human livers. The mRNA levels of *JUN*, *BMAL1* and *ELANE* were determined by qRT-PCR. Linear relationships between variables were tested using Pearson's correlation coefficient (n = 23). (F) Circadian neutrophil infiltration regulates hepatic metabolism through elastase, JNK and FGF21. Data are means ± SEM. *$p < 0.05$; **$p < 0.01$; (B) One-way ANOVA with Tukey's pots hoc test. (C) *t-test* or Welch's test.

The online version of this article includes the following source data for figure 5:

**Source data 1.** Baseline characteristics of the human cohort.

experiments, mice were intraperitoneally injected with SP600125 (15 mg/kg) (Santa Cruz Biotechnology) at ZT0. For neutrophil infusion experiments, mice were intravenously injected with $3 \times 10^6$ WT or NE$^{-/-}$ purified neutrophils each 3–4 days. Neutrophils were isolated from BM using biotinylated anti-Ly6G antibody (Clone:1A8) and streptavidin-labeled magnetic microbeads (Miltenyi Biotec).

All animal procedures conformed to EU Directive 86/609/EEC and Recommendation 2007/526/EC regarding the protection of animals used for experimental and other scientific purposes, enacted under Spanish law 1201/2005.

## Cell cultures

Hepatocytes were isolated from adult females by collagenase liver perfusion and cells were filtered through a 70 μm strainer. Hepatocytes pelleted from centrifuged Percoll gradients were plated at $4 \times 10^5$ cells/well on 6-well plates coated with collagen type one and incubated at 37°C. After 24 hr, cells were treated with 0.5 mM palmitate (Sigma-Aldrich) for 6 hr and then exposed for 1 hr to freshly neutrophils ($2 \times 10^6$ cells/well) in the presence of 1 μM FMLP (Sigma-Aldrich). Neutrophils were isolated from BM as described above. For some experiments, neutrophils were sorted purified form the BM using an anti-Ly6G antibody (Clone: 1A8). T and B lymphocytes were sorted purified from spleens using anti-CD3 (Clone: 145–2 C11) and anti-B220 (Clone: RA3-6B2), and bone marrow macrophages (BMDM) were differentiated as previously described (*González-Terán et al., 2013*). All antibodies were purchased from BD Pharmingen. Alternatively, hepatocytes were exposed 2 hr to 5 nM NE (R and D Systems) or 0.5 mg/mL of collagenase A (Roche) after palmitate treatment.

## Isolation of liver-infiltrating leukocytes

Mice were perfused with 20 mL of PBS and livers were collected and dissociated. Cell suspension was passed through a 70 μm strainer and centrifuged twice at 50 xg for 2 min to discard the liver parenchyma. For some experiments, livers were incubated for 15 min with 1 mg/mL Collagenase A (Roche) and 2 U/mL DNase (Sigma) at 37°C, and lungs were incubated for 25 min with 0,25 mg/ml Liberase TL (Sigma) and 5 U/mL DNase (Sigma) at 37°C Leukocyte fraction was collected and stained with anti-CD45 (Clone: 30-F11), from Invitrogen, anti-CD11b (Clone: M1/70), anti-Ly6G (Clone: 1A8) or anti-Ly6C/G (Clone: RB6-8C5), from BD Pharmingen, and alternatively, with anti-F4/80 (Clone: BM8), from Invitrogen, and Goat anti-Clec4F from R and D Systems and conjugated with anti-goat Alexa 647. Cells were sorted on a FACSAria to >95% purity. Flow cytometry experiments were performed with a FACScan cytofluorometer (FACS Canto BD), and data were analyzed with FlowJo software.

## Lentivirus vector production

Transient calcium phosphate transfection of HEK-293 cells (#CRL-1573, ATCC) was performed with the pGIPZ empty or pGIPZ.shFGF21 vector (V3LMM_430499 and V3LMM_430501, from Dharmacon) together with pΔ8.9 and pVSV-G. The supernatants were collected, centrifuged (700 xg, 4°C, 10 min) and concentrated (165x) by ultracentrifugation for 2 hr at 121,986 xg at 4°C (Ultraclear Tubes, SW28 rotor and Optima L-100 XP Ultracentrifuge; Beckman). Mice received tail-vein injections of 200 μl of lentiviral particles.

## RNA analysis

Expression of mRNA was examined by qRT-PCR using a 7900 Fast Real Time thermocycler and Fast Sybr Green assays (Applied Biosystems). Relative mRNA expression was normalized to *Gapdh* and *Actb* mRNA. The primers used were as follows: *Actb* (F: GGCTGTATTCCCCTCCATCG; R: CCAG

TTGGTAACAATGCCATGT); *Gapdh* (F: TGAAGCAGGGCATCTGAGGG; R: CGAAGGTGGAAGAG
TGGGA); *Clock* (F: AGAACTTGGCATTGAAGAGTCTC; R: GTCAGACCCAGAATCTTGGCT); *Bmal1*
(F: TGACCCTCATGGAAGGTTAGAA; R: GGACATTGCATTGCATGTTGG); *Nr1d2* (F: CAGACACTTC
TTAAAGCGGCACTG; R: GGAGTTCATGCTTGTGAAGGCTGT); *Cry2* (F: CACTGGTTCCGCAAAG-
GACTA; R: CCACGGGTCGAGGATGTAG); *Per2* (F: GAAAGCTGTCACCACCATAGAA; R: AAC
TCGCACTTCCTTTTCAGG); *Acaca* (F: GATGAACCATCTCCGTTGGC; R: GACCCAATTATGAA
TCGGGAGTG); *Fgf21* (F: CTGCTGGGGGTCTACCAAG; R: CTGCGCCTACCACTGTTCC); *Mip1a* (F:
TTCTCTGTACCATGACACTCTGC; R: CGTGGAATCTTCCGGCTGTAG); *Mip2* (F: CCAACCAC-
CAGGCTACAGG; R: GCGTCACACTCAAGCTCTG); *KC* (F: CTGGGATTCACCTCAAGAACATC; R:
CAGGGTCAAGGCAAGCCTC); *Sdf-1* (F: GCTCTGCATCAGTGACGGTA; R: ATCTGAAGGGCACAG
TTTGG); *Elane* (F: ATTTCCGGTCAGTGCAGGTAGT; R: GGTCAAAGCCATTCTCGAAGAT); *GAPDH*
(F: CCATGAGAAGTATGACAACAGCC; R: GGGTGCTAAGCAGTTGGTG); *ELANE* (F: TCCACGGAA
TTGCCTCCTTC; R: CCTCGGAGCGTTGGATGATA); *BMAL1* (F: GCCGAATGATTGCTGAGG; R:
CACTGGAAGGAATGTCTGG); *JUN* (F: GGATCAAGGCGGAGAGGAAG; R: GCGTTAGCATGAG
TTGGCAC).

## Measurement of hepatic triglycerides

Lipids were extracted from 25 mg of liver in isopropanol (50 mg/mL) and centrifuged (15 min 9500
xg 4°C). Triglycerides were detected in the supernatant (Sigma-Aldrich).

## Histology

Tissue samples were fixed in 10% formalin for 48 hr, dehydrated, and embedded in paraffin. Sections (5 μm) were cut and stained with hematoxylin and eosin (Sigma-Aldrich and Thermo Scientific).
Sections (8 μm) from frozen tissue and embedded in OCT compound (Tissue-Tek) were stained with
Oil Red O (American Master Tech Scientific). Sections were examined in Leica DM2500 microscope
using 20x objective.

## Immunoblotting

Tissue extracts were prepared in Triton lysis buffer [20 mM Tris (pH 7.4), 1% Triton X-100, 10% glycerol, 137 mM NaCl, 2 mM EDTA, 25 mM β-glycerophosphate, 1 mM sodium orthovanadate, 1 mM
phenylmethylsulfonyl fluoride, and 10 μg/mL aprotinin and leupeptin]. Extracts (20–50 μg protein)
were examined by immunoblot. The antibodies employed were anti-FGF21 (1/1000, #RD281108100,
BioVendor), anti-phospho JNK (1/1000, #4668S, Cell Signaling), anti-JNK (1/1000, #9252S, Cell Signaling), anti-phospho c-Jun (1/1000, #9164L, Cell Signaling), anti-c-Jun (1/1000, #9165S, Cell Signaling), anti-ACC (1/1000, #3676S, Cell Signaling), and anti-vinculin (1/5000, #V9131, Sigma). Anti-
phospho JNK and anti-JNK antibodies recognize the two different JNK isoform (JNK1 and JNK2)
and their two spliced variants (JNK1 (46 kDa), JNK1 (54 kDa) and JNK2 (46 kDa) and JNK2 (54 kDa)).
Immunocomplexes were detected by enhanced chemiluminescence (Amersham).

## Immunofluorescence

For 3-D imaging, livers were fixed in a solution of paraformaldehyde 4% in PBS at 4°C. After washing
in PBS, tissues were stored overnight in 30% sucrose (Sigma) with PBS. Then, livers were embedded
in OCT compound (Tissue-Tek) and frozen at −80°C. Cryosections of organs (70 μm) were washed in
PBS and blocked/permeabilized in PBS with 10% donkey serum (Millipore) and 1% Triton. Primary
antibodies diluted in blocking/permeabilization buffer were incubated overnight at 4°C, followed by
three washes in PBS and 2 hr incubation with secondary antibodies and DAPI at room temperature.
After three washes in PBS, cells were mounted with Fluoromount-G (SouthernBiotech). The following
primary and secondary antibodies were used: rat anti-CD31 (1:200, #553370 BD Pharmingen,), rabbit anti-S100A9 (mrp14) (1:100, #AB242945, Abcam,), goat anti-Clec4f (1:100, #AF2784, RD System),
Alexa 488 donkey anti rat IgG (1:200, #A-21208, ThermoFisher), Cy3 AffiniPure Fab Fragment Donkey Anti-Rabbit IgG (1:200, #711-167-003, Jackson Laboratories), Alexa Fluor 633 donkey anti goat
IgG (H+L) (1:200, #A21082, ThermoFisher). Immunostaining were imaged with a SP8 confocal microscope using 40x objectives. Individual fields or tiles of large areas were acquired every 2.5 μm for a
total of 30 μm in depth. 3D images were obtained with Fiji/ImageJ 3D Viewer plugging.

For 2-D imaging, liver sections (12 μm) prepared from frozen tissue and embedded in OCT compound were fixed with 2% paraformaldehyde and permeabilized with PBS 0.1% Triton. After blocking with PBS 5% BSA 0.1% Triton and washing, tissues were incubated overnight at 4°C with primary antibody. Then, sections were washed and incubated with conjugated secondary antibodies for 1 hr at room temperature and nuclei were stained with Sytox Green (Invitrogen) after washing. The following primary and secondary antibodies were used: rat anti-mouse S100A9 (Mrp-14) antibody (1:200, #AB105472, Abcam), rabbit anti-Neutrophil Elastase antibody (1:200, #AB68672, Abcam), goat Alexa Fluor 405 anti-rabbit (1:200) and goat Alexa Fluor 568 anti-rat IgG (1:500). Sections were mounted in Vectashield mounting medium (Vector, H-1000) and examined using a Leica SP5 multi-line inverted confocal microscope and 20x objectives.

## NE measurement

20 mL of PBS prefunded livers were crushed with a syringe plunger, resuspended in 4 mL of PBS/EDTA 5 mM/0.5% FBS and filtered (70 μm). Cell suspension was centrifuged at 1800 rpm 5 min and the supernatant was filtered (22 μm). Supernatants were concentrated using Amicon Ultra centrifugal filters (Sigma-Aldrich). NE levels were determined with Mouse Neutrophil Elastase ELISA kit (R and D system).

## Quantification and statistical analysis

All data are expressed as means ± SEM. For comparisons between two groups, the Student's *t*-test was applied. For data with more than two data sets, we used one-way ANOVA coupled with Turkey's multigroup test. When variances were unequal, Welch's test or Kruskal-Wallis test coupled with Dunn's multiple comparison test were applied, respectively. Multiple group comparisons in the rhythmicity of neutrophil infiltration were analyzed with two-way ANOVA followed by Fisher's *post hoc* test. Significance was determined as a 2-sided $p < 0.05$. All statistical analyses were conducted in GraphPad Prism software. Statistical details were indicated in the figure legends.

## Acknowledgements

We thank S Bartlett for English editing. We are grateful to A Zychlinsky for the NE[-/-] mice. We thank the staff at the CNIC Genomics, Cellomics, Microscopy, and Bioinformatics units for technical support and help with data analysis. BGT and MC were fellows of the FPI: Severo Ochoa CNIC program (SVP-2013–067639) and (BES-2017–079711) respectively. IN was funded by EFSD/Lilly grants (2017 and 2019), the CNIC IPP FP7 Marie Curie Programme (PCOFUND-2012–600396), EFSD Rising Star award (2019), JDC-2018-Incorporación (MIN/JDC1802). T-L was a Juan de la Cierva fellow (JCI-2011–11623). C.F has a Sara Borrell contract (CD19/00078). RJD is an Investigator of the Howard Hughes Medical Institute. This work was funded by the following grants to GS: funding from the European Union's Seventh Framework Programme (FP7/2007-2013) under grant agreement n° ERC 260464, EFSD/Lilly European Diabetes Research Programme Dr Sabio, 2017 Leonardo Grant for Researchers and Cultural Creators, BBVA Foundation (Investigadores-BBVA-2017) IN[17] _BBM_BAS_0066, MINECO-FEDER SAF2016-79126-R and PID2019-104399RB-I00 , EUIN2017-85875, Comunidad de Madrid IMMUNOTHERCAN-CM S2010/BMD-2326 and B2017/BMD-3733 and Fundación AECC AECC PROYE19047SABI and AECC: INVES20026LEIV to ML. MM was funded by ISCIII and FEDER PI16/01548 and Junta de Castilla y León GRS 1362/A/16 and INT/M/17/17 and JL-T by Junta de Castilla y León GRS 1356/A/16 and GRS 1587/A/17. The study was additionally funded by MEIC grants to ML (MINECO-FEDER-SAF2015-74112-JIN) AT-L (MINECO-FEDER-SAF2014-61233-JIN), RJD: Grant DK R01 DK107220 from the National Institutes of Health. AH: (SAF2015-65607-R). The CNIC is supported by the Instituto de Salud Carlos III (ISCIII), the Ministerio de Ciencia, Innovación y Universidades (MCNU) and the Pro CNIC Foundation, and is a Severo Ochoa Center of Excellence (SEV-2015–0505).

## Additional information

### Funding

| Funder | Grant reference number | Author |
| --- | --- | --- |
| European Commission | ERC260464 | Guadalupe Sabio |
| Ministerio de Economía y Competitividad | SAF2016-79126-R | Guadalupe Sabio |
| Ministerio de Economía y Competitividad | SAF2015-74112-JIN | Magdalena Leiva |
| Fundación Científica Asociación Española Contra el Cáncer | INVES20026LEIV PROYE19047SABI | Magdalena Leiva Guadalupe Sabio |
| Ministerio de Ciencia e Innovación | PID2019-104399RB-I00 | Guadalupe Sabio |
| FPI Severo Ochoa- CNIC | SVP-2013-067639 | Barbara Gonzalez-Teran |
| Ministerio de Economía y Competitividad | BES-2017-079711 | María Crespo |
| Juan de la Cierva | JCI-2011-11623 | Antonia Tomás-Loba |
| Sara Borrell | CD19/00078 | Cintia Folgueira |
| National Institutes of Health | DK R01 DK107220 | Roger J Davis |
| Ministerio de Economía y Competitividad | SAF2014-61233-JIN | Antonia Tomás-Loba |
| Fundación BBVA | IN[17]_BBM_BAS_0066 | Guadalupe Sabio |
| Ministerio de Economía y Competitividad | EUIN2017-85875 | Guadalupe Sabio |
| Comunidad de Madrid | S2010/BMD-2326 | Guadalupe Sabio |
| Comunidad de Madrid | B2017/BMD-3733 | Guadalupe Sabio |
| Instituto de Salud Carlos III | PI16/01548 | Miguel Marcos |
| Junta de Castilla y León | GRS1362/A/16 | Miguel Marcos |
| Junta de Castilla y León | INT/M/17/17 | Miguel Marcos |
| Junta de Castilla y León | GRS 1356/A/16 | Jorge L Torres |
| Junta de Castilla y León | GRS 1587/A/17 | Jorge L Torres |
| European Foundation for the Study of Diabetes | ESFD/Lilly Programme | Guadalupe Sabio |
| European Foundation for the Study of Diabetes | EFSD/Lilly Grant 2017 and 2019 | Ivana Nikolic |
| CNIC IPP FP7 Marie Curie Programme | PCOFUND-2012-600396 | Ivana Nikolic |
| European Foundation for the Study of Diabetes | EFSD Rising Star award 2019 | Ivana Nikolic |
| Juan de la Cierva | JDC-2018-Incorporación MIN/JDC1802 | Ivana Nikolic |

The funders had no role in study design, data collection and interpretation, or the decision to submit the work for publication.

### Author contributions

María Crespo, Barbara Gonzalez-Teran, Data curation, Formal analysis, Investigation, Methodology, Writing - review and editing; Ivana Nikolic, Cintia Folgueira, Macarena Fernández-Chacón, Antonia Tomás-Loba, Noelia A-Gonzalez, Daniel Beiroa, Formal analysis, Investigation, Methodology; Alfonso Mora, Formal analysis, Investigation, Methodology, Writing - review and editing; Elena

Rodríguez, Luis Leiva-Vega, Aránzazu Pintor-Chocano, Ainoa Caballero-Molano, Lourdes Hernández-Cosido, Jorge L Torres, Norman J Kennedy, Roger J Davis, Rui Benedito, Miguel Marcos, Ruben Nogueiras, Andrés Hidalgo, Formal analysis, Methodology; Irene Ruiz-Garrido, Beatriz Cicuéndez, Data curation, Methodology; Nuria Matesanz, Magdalena Leiva, Conceptualization, Data curation, Formal analysis, Supervision, Investigation, Methodology, Writing - original draft, Writing - review and editing; Guadalupe Sabio, Conceptualization, Data curation, Supervision, Funding acquisition, Investigation, Methodology, Writing - original draft, Project administration, Writing - review and editing

Author ORCIDs
María Crespo ⓘ https://orcid.org/0000-0003-3666-3415
Barbara Gonzalez-Teran ⓘ https://orcid.org/0000-0002-4336-8644
Alfonso Mora ⓘ https://orcid.org/0000-0002-6397-4836
Noelia A-Gonzalez ⓘ http://orcid.org/0000-0003-0533-5216
Roger J Davis ⓘ http://orcid.org/0000-0002-0130-1652
Magdalena Leiva ⓘ https://orcid.org/0000-0001-7735-2459
Guadalupe Sabio ⓘ https://orcid.org/0000-0002-2822-0625

Ethics
Human subjects: The study was approved by the Ethics Committee of the University Hospital of Salamanca (Spain), and all subjects provided written informed consent to participate.
Animal experimentation: All animal procedures conformed to EU Directive 86/609/EEC and Recommendation 2007/526/EC regarding the protection of animals used for experimental and other scientific purposes, enacted under Spanish law 1201/2005.

Decision letter and Author response
Decision letter https://doi.org/10.7554/eLife.59258.sa1
Author response https://doi.org/10.7554/eLife.59258.sa2

## Additional files

### Supplementary files
• Transparent reporting form

### Data availability
All data generated or analysed during this study are included in the manuscript and supporting files.

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

# Appendix 1

**Appendix 1—key resources table**

| Reagent type (species) or resource | Designation | Source or reference | Identifiers | Additional information |
|---|---|---|---|---|
| Genetic reagent (*M. musculus*) | C57BL/6J background | Jackson Laboratory | Cat# 000664 RRID:IMSR_ JAX:000664 | |
| Genetic reagent (*M. musculus*) | B6.129-Mcl1tm3Sjk/J | Jackson Laboratory | Cat# 006088 RRID:IMSR_ JAX:006088 | |
| Genetic reagent (*M. musculus*) | B6.Cg-Tg(S100A8-cre,-EGFP)1Ilw/J | Jackson Laboratory | Cat# 021614 RRID:IMSR_ JAX:021614 | |
| Genetic reagent (*M. musculus*) | B6.129P2-Lyz2tm1(cre)Ifo/J | Jackson Laboratory | Cat# 004781 RRID:IMSR_ JAX:004781 | |
| Genetic reagent (*M. musculus*) | B6.129-Mapk12tm1.2 | PMID:26843485 | | |
| Genetic reagent (*M. musculus*) | B6.129-Mapk13tm1.2 | PMID:26843485 | | |
| Genetic reagent (*M. musculus*) | B6.129 × 1/SvJ-Elanetm1Sds | Jackson Laboratory | Cat# 006112 RRID:IMSR_ JAX:006112 | |
| Genetic reagent (*M. musculus*) | B6.Cg-Tg(Alb-cre)21Mgn/J | Jackson Laboratory | Cat# 003574 RRID:IMSR_ JAX:003574 | |
| Genetic reagent (*M. musculus*) | B6.129-Mapk8LoxP/LoxP Mapk9tm1Flv/J | PMID:19167327 | | |
| Genetic reagent (*M. musculus*) | C57BL/6-Cxcr2tm1Rmra/J | Jackson Laboratory | Cat# 024638 RRID:IMSR_ JAX:024638 | |
| Cell line (*H. sapiens*) | HEK-293 | ATCC | Cat# CRL-1573 RRID:CVCL_ 0045 | |
| Cell line (*M. musculus*) | Primary hepatocytes | PMID:26843485 | | |
| Transfected construct (synthesized) | pGIZP (pΔ8.9- pVSV-G) | Dharmacon | Cat# RHS4349 | Lentiviral Empty Vector shRNA Control |
| Transfected construct (synthesized) | pGIZP.shFGF21 (pΔ8.9-pVSV-G) | Dharmacon | Cat# V3LMM_430499 | |
| Transfected construct (synthesized) | pGIZP.shFGF21 (pΔ8.9-pVSV-G) | Dharmacon | Cat# V3LMM_430501 | |
| Biological sample (*H. sapiens*) | Liver human samples | University Hospital of Salamanca-IBSAL | *Figure 5— source data 1* | |
| Antibody | Biotinylated monoclonal rat anti-mouse Ly6G (Clone 1A8) | Miltenyi Biotec | Cat# 130-123-854 RRID:AB_ 1036098 | 1:20 |
| Antibody | Biotinylated monoclonal hamster anti-mouse CD3 (Clone 145–2 C11) | BD Pharmingen | Cat# 553057 RRID:AB_ 394590 | 1:20 |

*Continued on next page*

*Appendix 1—key resources table continued*

| Reagent type (species) or resource | Designation | Source or reference | Identifiers | Additional information |
|---|---|---|---|---|
| Antibody | Biotinylated monoclonal rat anti-mouse B220 (Clone RA3-6B2) | BD Pharmingen | Cat# 561880 RRID:AB_10897020 | 1:20 |
| Antibody | Monoclonal rat anti-mouse CD45 Pacific Orange (Clone 30-F11) | Invitrogen | Cat# MCD4530 RRID:AB_2539700 | Flow cytometry 1:100 |
| Antibody | Monoclonal rat anti-mouse CD11b FITC (Clone M1/70) | BD Pharmingen | Cat# 557396 RRID:AB_396679 | Flow cytometry 1:100 |
| Antibody | Monoclonal rat anti-mouse Ly6C/G APC (Clone RB6-8C5) | BD Pharmingen | Cat# 553129 RRID:AB_398532 | Flow cytometry 1:200 |
| Antibody | Monoclonal rat anti-mouse F4/80 PE-Cy7 (Clone BM8) | eBioscience | Cat# 25480182 RRID:AB_469653 | Flow cytometry 1:100 |
| Antibody | Monoclonal rat anti-Mouse Ly-6G PE (Clone 1A8) | BD Bioscience | Cat# 551461 RRID:AB_394208 | Flow cytometry 1:200 |
| Antibody | Polyclonal Chicken Anti Goat IgG (H+L) Alexa Fluor 647 | Invitrogen | Cat# A-21469 RRID:AB_2535872 | Flow cytometry 1:500 |
| Antibody | Polyclonal rabbit anti-mouse FGF21 | BioVendor | Cat# RD281108100 RRID:AB_2034054 | WB 1:1000 |
| Antibody | Monolconal rabbit anti-phospho SAPK/JNK (T183/Y185) (Clone 81E11) | Cell Signaling | Cat# 4668S RRID:AB_823588 | WB 1:1000 |
| Antibody | Polyclonal rabbit anti-SAPK/JNK | Cell Signaling | Cat# 9252S RRID:AB_2250373 | WB 1:1000 |
| Antibody | Polyclonal rabbit anti-phospho c-jun | Cell Signaling | Cat# 9164L RRID:AB_330892 | WB 1:1000 |
| Antibody | Monoclonal rabbit anti-c-jun (Clone 60A8) | Cell Signaling | Cat# 9165S RRID:AB_2130165 | WB 1:1000 |
| Antibody | Monoclonal rabbit anti-Acetyl-CoA carboxylase (Clone C83B10) | Cell Signaling | Cat# 3676S RRID:AB_2219397 | WB 1:1000 |
| Antibody | Monoclonal mouse anti-vinculin (Clone hVIN-1) | Sigma | Cat# V9131 RRID:AB_477629 | WB 1:5000 |
| Antibody | Polyclonal goat anti-Mouse IgG (H+L) Secondary Antibody, HRP | ThermoFisher | Cat# 31430 RRID:AB_228307 | WB 1:5000 |
| Antibody | Polyclonal goat anti-Rabbit IgG (H+L) Secondary Antibody, HRP | ThermoFisher | Cat# 31460 RRID:AB_228341 | WB 1:5000 |
| Antibody | Monoclonal rat anti-mouse CD31 (Clone MEC 13.3) | BD Pharmingen | Cat# 553370 RRID:AB_394816 | IF 1:200 |

*Continued on next page*

*Appendix 1—key resources table continued*

| Reagent type (species) or resource | Designation | Source or reference | Identifiers | Additional information |
|---|---|---|---|---|
| Antibody | Monoclonal rabbit anti-mouse S100A9 (mrp14) (Clone EPR22332-75) | Abcam | Cat# AB242945 RRID:AB_2876886 | IF 1:100 |
| Antibody | Polyclonal goat anti-mouse Clec4f | RD System | Cat# AF2784 RRID:AB_2081339 | IF/Flow cytometry 1:200 |
| Antibody | Polyclonal donkey anti rat IgG Alexa 488 | ThermoFisher | Cat# A-21208 RRID:AB_2535794 | IF 1:200 |
| Antibody | Polyclonal Donkey Anti-Rabbit IgG Cy3 AffiniPure Fab Fragment | Jackson Laboratories | Cat# 711-167-003 RRID:AB_2340606 | IF 1:200 |
| Antibody | Polyclonal Donkey Anti Goat IgG (H+L) Alexa Fluor 633 | ThermoFisher | Cat# A21082 RRID:AB_10562400 | IF 1:200 |
| Antibody | Monoclonal rat anti-mouse S100A9 (Mrp-14) (Clone 2B10) | Abcam | Cat# AB105472 RRID:AB_10862594 | IF 1:200 |
| Antibody | Polyclonal rabbit anti-neutrophil elastase | Abcam | Cat# AB68672 RRID:AB_1658868 | IF 1:200 |
| Antibody | Polyclonal goat Anti-Rabbit Alexa Fluor 405 | ThermoFisher | Cat# A-31556 RRID:AB_221605 | IF 1:200 |
| Antibody | Polyclonal goat Anti-Rat IgG Alexa Fluor 568 | ThermoFisher | Cat# A-11077 RRID:AB_2534121 | IF 1:500 |
| Sequence-based reagent | RT-qPCR primers | Sigma-Aldrich | | |
| Peptide, recombinant protein | Recombinant Mouse Neutrophil Elastase/EL | R and D Systems | Cat# 4517-SE-010 | |
| Peptide, recombinant protein | Collagenase A | Roche | Cat# 10 103 586 001 | |
| Peptide, recombinant protein | Collagenase Type 1 CLS1 | Worthington Biochemical | Cat# LS004197 | |
| Peptide, recombinant protein | Liberase TL | Sigma | Cat# 5401020001 | |
| Peptide, recombinant protein | DNase Type II-S | Sigma-Aldrich | Cat# D4513 | |
| Commercial assay or kit | Serum Triglyceride Determination Kit | Sigma-Aldrich | Cat# TR0100-1KT | |
| Commercial assay or kit | Mouse Neutrophil Elastase/ ELA2 DuoSet ELISA | R and D systems | Cat# DY4517-05 | |
| Commercial assay or kit | RNa easy Mini Kit | Qiagen | Cat# 74106 | |

*Continued on next page*

*Appendix 1—key resources table continued*

| Reagent type (species) or resource | Designation | Source or reference | Identifiers | Additional information |
|---|---|---|---|---|
| Commercial assay or kit | High-Capacity cDNA Reverse Transcription Kit | Applied Biosystems | Cat# 4368814 | |
| Chemical compound, drug | Fast SYBR Green Master Mix | Applied Biosystems | Cat# 4385616 | |
| Chemical compound, drug | Percoll | GE Healthcare | Cat# 17-0891-01 | |
| Chemical compound, drug | Palmitic acid | Sigma-Aldrich | Cat# P0500 | |
| Chemical compound, drug | N-Formil Met-Leu-Phe (FMLP) | Sigma-Aldrich | Cat# F3506 | |
| Chemical compound, drug | SP600125 (SAPK inhibitor) | Santa Cruz Biotechnology | Cat# sc-200635 | |
| Chemical compound, drug | Amersham ECL Prime Western Blotting Detection Reagent | GE Healthcare | Cat# RPN2232 | |
| Chemical compound, drug | Fluoromount-G | SouthernBiotech | Cat# 0100–01 | |
| Chemical compound, drug | Sucrose | Sigma-Aldrich | Cat# S8501 | |
| Chemical compound, drug | SYTOX Green Nucleic Acid Stain - 5 mM | ThermoFisher | Cat# S7020 | |
| Chemical compound, drug | VECTASHIELD Antifade Mounting Medium | Vector Lab | Cat# H-1000 | |
| Software, algorithm | GraphPad PRISM | GraphPad Software | RRID:SCR_002798 | |
| Software, algorithm | Photoshop CS6 | Adobe | RRID:SCR_014199 | |
| Software, algorithm | Fiji/Image J software | Fiji-Image J | https://imagej.nih.gov/ij/ RRID:SCR_003070 | |
| Software, algorithm | FlowJo | FlowJo | https://www.flowjo.com/ RRID:SCR_008520 | |
| Software, algorithm | Leica LAS X | Leica Software | RRID:SCR_013673 | |
| Other | Hematoxylin | Sigma | Cat# H3136 | |
| Other | Eosin Y Alcoholic | Thermo Scientific | Cat# 6766008 | |
| Other | OCT | Tissue-Tek | Cat# 4583 | |
| Other | Oil Red O (C.I.26125) | American Master Tech Scientific | Cat# SPO1077 | |
| Other | 70 µM cell strainers | Corning Falcon | Cat# 352350 | |
| Other | 22 µM filter | Sigma-Aldrich | Cat# SLGPM33RS | |
| Other | Amicon Ultra centrifugal filters | Sigma-Aldrich | Cat# UFC800324 | |
| Other | Magnetic streptavidin microbeads | Miltenyi Biotec | Cat# 130-048-101 | |

*Continued on next page*

*Appendix 1—key resources table continued*

| Reagent type (species) or resource | Designation | Source or reference | Identifiers | Additional information |
|---|---|---|---|---|
| Other | MACS Separation Columns- MS columns | Miltenyi Biotec | Cat# 130-042-201 | |
| Other | Mini-osmotic pumps | Alzet | Cat# 1004 | |
| Other | Methionine-choline-deficient diet (MCD) | Research Diets Inc | Cat# A02082002B | |
| Other | High-fat diet (HFD) | Research Diets Inc | Cat# D11103002i | |

