## [Decision Letter]

**Acceptance summary:**

This study investigating how neutrophils regulate daily hepatic homeostasis, showed that in steady state, neutrophils infiltrate the liver following a circadian pattern and regulate hepatocyte clock-genes by neutrophil elastase (NE) secretion and more mechanistically through a NE/*JNK*/*Bmal1* axis as well as the hepatokine FGF21. They also show that dysregulation of such circadian neutrophil infiltration alters clock genes expression leading to rise in triglyceride content in the liver.

**Decision letter after peer review:**

Thank you for submitting your article "Neutrophil infiltration regulates clock gene expression to organize daily hepatic metabolism" for consideration by *eLife*. Your article has been reviewed by three peer reviewers, one of whom is a member of our Board of Reviewing Editors, and the evaluation has been overseen by Carla Rothlin as the Senior Editor. The reviewers have opted to remain anonymous.

The reviewers have discussed the reviews with one another and the Reviewing Editor has drafted this decision to help you prepare a revised submission.

Summary:

The authors investigated here how neutrophils regulate daily hepatic homeostasis. They show that in steady state, neutrophils infiltrate the liver following a circadian pattern and regulate hepatocyte clock-genes by neutrophil elastase (NE) secretion and more mechanistically thru a NE/*JNK*/*Bmal1* axis as well as the hepatokine FGF21. They also show that dysregulation of such circadian neutrophil infiltration alters clock genes expression leading to rise in triglyceride content in the liver. Finally, they show that such neutrophil infiltration and clock genes expression are modulated by β3-adrenergic receptor signalling. This is overall a well-designed and performed study with novelty that will be of interest for the community.

Essential revisions:

All three reviewers agree on these four important points:

1) The authors use the Lyzs-Cre model to remove genes in neutrophils. However, such model is not strictly neutrophil-specific and will affect monocyte and macrophages including Kupffer cells, the liver resident macrophage population (See Greenhalgh et al. 2015, Hepatology). They should exclude any possible contribution of the cells to their findings.

2) More in vivo evidence on the role of neutrophils: The control of circadian clock expression is manifold and highly complex. While the authors do show a correlation between higher neutrophil number in the liver and higher *Bmal1* expression, the “control” effect of neutrophils on *Bmal1* expression is not fully convincing. In neutropenic mice, expression levels of *Bmal1* still go up, although neutrophils are reduced. Overall *Bmal1* levels seem to be very similar between the two models. How can neutropenic mice after jetlag rescue the control jet lag phenotype with respect to *Bmal1* expression if neutrophils were important in controlling liver clock gene expression per se?

A better model might be not to target neutrophils directly but to block rhythmic migration to the liver, e.g. with antibodies directed against VCAM-1.

Furthermore, can the neutropenic phenotype be rescued with the infusion of NE+ but not NE- neutrophils?

3) The β 3 antagonist treatment feels out of place. What was the rationale to use this treatment and why was not the β 2 adrenergic receptor chosen, the one that is actually expressed in neutrophils? This aspect of the paper raises more questions than it aims to answer – given also the effects of the sympathetic nervous system as an entrainer for circadian rhythms in peripheral tissues – and should probably be best left out.

4) For the HFD and MCD experiments, the authors should perhaps provide the rationale why 2 different models were used interchangeably or should just stick with one model. Figure 2, for non-expert, the experimental protocol for the jetlag is not well defined, making it difficult to interpret the data. It is unclear what is the rationale for the authors to interchange between the use of HFD and MCD diet? Perhaps the authors should clearly state that what they aim to understand from the use of each diet.

---

## [Author Response]

Essential revisions:All three reviewers agree on these four important points:1) The authors use the Lyzs-Cre model to remove genes in neutrophils. However, such model is not strictly neutrophil-specific and will affect monocyte and macrophages including Kupffer cells, the liver resident macrophage population (See Greenhalgh et al. 2015, Hepatology). They should exclude any possible contribution of the cells to their findings.

We thank the reviewer for her/his encouraging comment and we understand their concern. However, Dzhagalov et al. previously characterized the myeloid compartment of the MCL1^Lyzs‑KO^ mice and they demonstrated that the deletion of MCl^-^1 mainly disturbs the neutrophil compartment without affecting the macrophage population and their functionality^1^. Additionally, we have characterized the distinct myeloid populations in the liver and the bone marrow without detecting any important changes in non-neutrophil populations (Figure 1—figure supplement 1E and Figure 1—figure supplement 2C 1).

In addition, we have used the Mrp8-Cre model to remove Mcl1 in neutrophils in steady state and we found similar results to the Lyzs-Cre model (Figure 1G).

However, deletion of Mcl1 under Mrp8-Cre promoter is deleterious for mice resulting in smaller (Author response image 1) and non-fertile mice, and some of them die early after birth ^2^. Indeed, it has been described the expression of Mrp8 in the cytotrophoblast, placental-tissue macrophages and fibroblasts during embryonic development^3^, while Mcl1 has demonstrated to be essential for germinal-derived cells such as oocytes^4^. It is not surprising then that deletion of Mcl1 using the Mrp8-Cre system has detrimental effects over other tissues. To avoid these problems, we performed bone marrow transplantation, but, unfortunately, bone marrow of MCL1^Mrp8-KO^ mice was not able to properly reconstitute the animals, since mice died after the transplant, in agreement with a previous report ^2^. Therefore, Mrp8-Cre model was avoided and Lyzs-Cre model used in the rest of the experiments.

**Author response image 1. sa2fig1:** Representative images of MRP8-Cre and MCL^-^1MRP8-KO mice. 8 weeks-old Mrp8-Cre (size: 8cm, weight: 23.36g) and MCL1Mrp8-KO (size: 6.8cm, weight: 17.3g) male mice and their size are shown.

As we agree with the reviewer that demonstration of neutrophil specificity is an important point, we used mice lacking *Cxcr2* in neutrophils using Mrp8-Cre promoter as another model to specifically reduce neutrophil migration and therefore, as a model to evaluate neutrophil-hepatocyte communication. *Cxcr2* has been shown to be essential for neutrophil recruitment from the bone marrow to tissues and in consequence, mice lacking *Cxcr2* presented impaired neutrophil tissue infiltration^5,6^. Using this model, we evaluated the effect of reduction of neutrophil infiltration in liver in the model of jet lag. We found that lack of *Cxcr2* specifically in neutrophils reduced *Bmal1* expression and triglycerides content in the liver (Figure 2—figure supplement 1C,D) to the same extent as MCL1^Lyzs-KO^ mice (Figure 2C,D). These results corroborate the specific effect of neutrophils in the regulation of liver circadian clock.

2) More in vivo evidence on the role of neutrophils: The control of circadian clock expression is manifold and highly complex. While the authors do show a correlation between higher neutrophil number in the liver and higher Bmal1 expression, the “control” effect of neutrophils on Bmal1 expression is not fully convincing. In neutropenic mice, expression levels of Bmal1 still go up, although neutrophils are reduced. Overall Bmal1 levels seem to be very similar between the two models. How can neutropenic mice after jetlag rescue the control jet lag phenotype with respect to Bmal1 expression if neutrophils were important in controlling liver clock gene expression per se?

The effects in homeostasis are always subtle as *Bmal1* might be controlled not only by neutrophils. However, even that is true, the lack of neutrophils results in a significant reduction of *Bmal1*. Our hypothesis is that neutrophils infiltration has a circadian rhythm that induces *Bmal1* expression and, in consequence, lipogenesis. Under jet lag, the circadian rhythmicity of neutrophil infiltration is lost and neutrophils are infiltrated during day and night, inducing constantly *Bmal1* expression and lipogenesis and, in consequence, steatosis. Thus, under conditions where there is an abrupt pattern of neutrophil infiltration in the liver, lack of neutrophils protects from the increase in *Bmal1* expression and, hence, from increased lipogenesis and steatosis.

A better model might be not to target neutrophils directly but to block rhythmic migration to the liver, e.g. with antibodies directed against VCAM-1.

We agree with the reviewer that this model could nicely strength our findings. For this reason, we decided to use the Mrp8 Cxcr2 KO mice. As indicated above, Cxcr2 has been shown to be essential for the recruitment of neutrophils from the bone marrow to tissues and, therefore, mice lacking Cxcr2 present impaired infiltration of neutrophil in tissues^5,6^. As we show in Figure 2—figure supplement 1C,D , the deletion of Cxcr2 in neutrophils is enough to reduce *Bmal1* expression and triglycerides content in the liver after jet lag protocol, indicating that lack of migration of neutrophils to the liver affects hepatic circadian clock.

To further evaluate the effect of neutrophils migration, we used a second model, p38γ/δ ^LyzKO^ mice. Lack of p38γ/δ in neutrophils impairs neutrophil migration and infiltration in the liver^7^. We corroborated that these animals presented significant lower levels of infiltrated neutrophils in the liver after jet lag. This reduction in neutrophils infiltration correlated with lower levels of hepatic *Bmal1* expression and reduced steatosis after jet lag (Figure 2—figure supplement 1E, F,G).

Furthermore, can the neutropenic phenotype be rescued with the infusion of NE+ but not NE- neutrophils?

We thank the reviewer for this suggestion. We performed the experiment using neutropenic mice (MCL1^Lyzs-KO^) without infusion, and infused with WT neutrophils or Elastase KO neutrophils (NE^-/-^). The infusion of WT neutrophils was able to increase *Bmal1* expression in the liver after jet lag, while neutropenic mice infused with NE^-/-^ neutrophils presented the same levels of *Bmal1* than neutropenic mice. In addition, while infusion of neutropenic mice with WT neutrophils increased steatosis, neutropenic mice infused with NE^-/-^ neutrophils presented the same levels of steatosis than neutropenic mice non-infused. These new data indicate that neutrophil elastase plays an essential role in the control of circadian clock in the liver (Figure 5A-D).

3) The β 3 antagonist treatment feels out of place. What was the rationale to use this treatment and why was not the β 2 adrenergic receptor chosen, the one that is actually expressed in neutrophils? This aspect of the paper raises more questions than it aims to answer – given also the effects of the sympathetic nervous system as an entrainer for circadian rhythms in peripheral tissues – and should probably be best left out.

We have followed the recommendation of the reviewer and left out this part of the paper.

4) For the HFD and MCD experiments, the authors should perhaps provide the rationale why 2 different models were used interchangeably or should just stick with one model. Figure 2, for non-expert, the experimental protocol for the jetlag is not well defined, making it difficult to interpret the data. It is unclear what is the rationale for the authors to interchange between the use of HFD and MCD diet? Perhaps the authors should clearly state that what they aim to understand from the use of each diet.

MCD diet lacks methionine and choline, which are indispensable for hepatic mitochondrial β-oxidation and very low-density lipoprotein (VLDL) synthesis^8^. This diet induces steatohepatitis, necroinflammation, and fibrosis similar to human NASH, and it is considered one of the best-established models for studying NASH-associated inflammation, oxidative stress, and fibrosis. We used this model in mostly all the paper because with this model we observed a high increase of neutrophil infiltration in the liver. In this model, the steatosis and the increase in triglycerides in the liver are independent from obesity. But, on the other hand, we also wanted to evaluate whether the effect that we observed in the MCD diet also appears in a model of obesity-induced steatosis, using, for this reason, HFD.

**References**

1 Dzhagalov, I., St John, A. & He, Y. W. The antiapoptotic protein MCl^-^1 is essential for the survival of neutrophils but not macrophages. *Blood* 109, 1620-1626, doi:10.1182/blood-2006-03-013771 (2007).2 Csepregi, J. Z. *et al.* Myeloid-Specific Deletion of MCl^-^1 Yields Severely Neutropenic Mice That Survive and Breed in Homozygous Form. *Journal of immunology* 201, 3793-3803, doi:10.4049/jimmunol.1701803 (2018).3 Sato, N., Isono, K., Ishiwata, I., Nakai, M. & Kami, K. Tissue expression of the S100 protein family-related MRP8 gene in human chorionic villi by in situ hybridization techniques. *Okajimas Folia Anat Jpn* 76, 123-129, doi:10.2535/ofaj1936.76.2-3_123 (1999).4 Omari, S. *et al.* MCl^-^1 is a key regulator of the ovarian reserve. *Cell Death Dis* 6, e1755, doi:10.1038/cddis.2015.95 (2015).5 Mei, J. *et al.* Cxcr2 and Cxcl5 regulate the IL-17/G-CSF axis and neutrophil homeostasis in mice. *The Journal of clinical investigation* 122, 974-986, doi:10.1172/JCI60588 (2012).6 Eash, K. J., Greenbaum, A. M., Gopalan, P. K. & Link, D. C. CXCR2 and CXCR4 antagonistically regulate neutrophil trafficking from murine bone marrow. *The Journal of clinical investigation* 120, 2423-2431, doi:10.1172/JCI41649 (2010).7 Gonzalez-Teran, B. *et al.* p38gamma and p38delta reprogram liver metabolism by modulating neutrophil infiltration. *The EMBO journal* 35, 536-552, doi:10.15252/embj.201591857 (2016).8 Anstee, Q. M. & Goldin, R. D. Mouse models in non-alcoholic fatty liver disease and steatohepatitis research. *International journal of experimental pathology* 87, 1-16, doi:10.1111/j.0959-9673.2006.00465.x (2006).